# Multidimensional Results and Reflections on CAR-T: The Italian Evidence

**DOI:** 10.3390/ijerph20053830

**Published:** 2023-02-21

**Authors:** Emanuela Foglia, Elisabetta Garagiola, Vito Ladisa, Alessandro Rambaldi, Roberto Cairoli, Simona Sammassimo, Emanuela Omodeo Salè, Pier Luigi Zinzani, Marco Esposti, Luisa Alberti, Maria Franca Mulas, Eleonora Melis, Stefania Onnis, Maurizio Marcias, Vittorio Satta, Davide Croce

**Affiliations:** 1Centre for Research on Health Economics, Social and Health Care Management, LIUC-Università Cattaneo, 21053 Castellanza, Italy; 2Hospital Pharmacy, IRCCS National Cancer Institute Foundation, 20133 Milan, Italy; 3Department of Oncology and Hematology, Papa Giovanni XXIII Hospital, 24127 Bergamo, Italy; 4Division of Hematology, Grande Ospedale Metropolitano Niguarda Hospital, 20162 Milano, Italy; 5Department of Oncology and Hematology-Oncology, European Institute of Oncology, 20141 Milan, Italy; 6Hospital Pharmacy, European Institute of Oncology, 20141 Milan, Italy; 7Institute of Hematology “Seragnoli”, IRCCS University Hospital of Bologna, 40139 Bologna, Italy; 8Department of Specialized, Diagnostic and Experimental Medicine, University of Bologna, 40139 Bologna, Italy; 9Management Control, Lodi Hospital, 26900 Lodi, Italy; 10Territorial Pharmaceutical Complex Structure, Regional Health Authority—ARES Sardinia, 09047 Cagliari, Italy; 11General Direction, Policlinico Tor Vergata, 00133 Rome, Italy; 12Complex Structure for Planning and Management Control, Regional Health Authority—ARES Sardinia, 07100 Sassari, Italy; 13Territorial District 3—Quartu Parteolla, Local Healthcare Authority—ASL 8, 09126 Cagliari, Italy; 14Complex Structure of Pharmacoeconomics and Pharmacovigilance, Regional Health Authority—ARES Sardinia, 09047 Cagliari, Italy; 15Complex Structure Health Technology Assessment, Regional Health Authority—ARES Sardinia, 09047 Cagliari, Italy

**Keywords:** diffuse large B-cell lymphoma, CAR-T cells, Best Salvage Care, HTA, organizational impact, economic sustainability, Italy

## Abstract

The present study aims at defining the economic and organizational impacts of the introduction of chimeric antigen receptor T-cell therapy (CAR-T) in Italy, for the management of diffuse large B-cell lymphoma (DLBCL) patients in third-line therapy, defining the overall level of sustainability for both hospitals and the National Healthcare System (NHS). The analysis focused on CAR-T and Best Salvage Care (in the following BSC), assuming the Italian hospital and NHS perspectives, over a 36-month time horizon. Process mapping and activity-based costing methodologies were applied to collect the hospital costs related to the BSC and CAR-T pathways, including adverse event management. Anonymous administrative data on services provided (diagnostic and laboratory examinations, hospitalizations, outpatient procedures, and therapies) to 47 third-line patients with lymphoma, as well as any organizational investments required, were collected, in two different Italian Hospitals. The economic results showed that the BSC clinical pathway required less resources in comparison with CAR-T (excluding the cost related to the therapy) (BSC: 29,558.41 vs. CAR-T: EUR 71,220.84, −58.5%). The budget impact analysis depicts that the introduction of CAR-T would generate an increase in costs ranging from 15% to 23%, without considering treatment costs. The assessment of the organizational impact reveals that the introduction of CAR-T therapy would require additional investments equal to a minimum of EUR 15,500 to a maximum of EUR 100,897.49, from the hospital perspective. Results show new economic evidence for healthcare decision makers, to optimize the appropriateness of resource allocation. The present analysis suggests the need to introduce a specific reimbursement tariff, both at the hospital and at NHS levels, since no consensus exists, at least in the Italian setting, concerning the proper remuneration for the hospitals who guarantee this innovative pathway, assuming high risks related to timely management of adverse events.

## 1. Introduction

Cancer imposes a major disease burden worldwide, with considerable geographic variations in incidence, mortality, and survival rates, as well as differences in prevention, detection, treatment, and palliative programs. The treatment of these conditions often requires long-term monitoring, with the development of a personalized approach based on clinical settings. Public health interventions need to consider the changes occurring and modify the approaches as well as treatments available for the proper management and delivery of services of this pathology.

Among all the different types of cancers, diffuse large B-cell lymphoma (DLBCL) is an aggressive pathology, accounting for approximately 30% of all lymphomas [1,2].

The standard of care for the treatment of DLBCL is chemo-immunotherapy (particularly with R-CHOP -rituximab, cyclophosphamide, doxorubicin, vincristine, and prednisone), a strategy that is safe and effective, although often responsible for severe side effects [3]. Despite its continuous refinement, a high percentage of treatment failure and refractory cases are registered, with a poor prognosis [3,4,5]. Empirical combination chemotherapy cures approximately 65% of patients initially, with another 20% to 25% cured with salvage therapies [2]. Patients who remain refractory to treatment will have a median overall survival (OS) of a maximum of 10 months, indicating the presence of significant unmet medical needs, for high-risk groups [3]. Refractory patients could undergo an autologous steam cell transplant, but more than 60% of patients are not eligible for such a procedure [6].

In this view, there are limited treatment options and emergent substantial unmet medical needs for adult patients with relapsed or refractory DLBCL, generating a significant challenge for healthcare systems worldwide.

The development of novel and more effective and promising therapies is thus required for this population: targeted therapies and combinations, as well as cellular and immunotherapeutic agents, are still under investigation. Among these, the most promising therapy is represented by chimeric antigen receptor T-cell therapy, known as CAR-T cells or CAR-T [7]. CAR-T therapies are potentially curative treatments and thus ground-breaking. These innovative therapies genetically engineer patients’ blood cells to target tumors [8]. Indeed, axicabtagene ciloleucel, tisagenlecleucel and lisocabtagene maraleucel are the currently FDA-approved CAR T-cell products, for the treatment of relapsed or refractory DLBCL, after the second lines of systemic therapy.

The clinical relevance of CAR-T in the treatment of DLBCL is fully recognized by different stakeholders, revolutionizing the outcomes of refractory patients affected by this disease. The relative approval studies (the ZUMA-1 trial for axi-cel [9], the JULIET trial for tisa-cel [10] and the phase 1 TRANSCEND NHL 001 trial [11] reported a high Overall Response Rate (ORR), ranging from 52% to 83%, and a Complete Response (CR), ranging from 40% to 54%. In addition to the great efficacy, the therapy is well tolerated by patients, although the toxicity profile of the innovative therapy could be associated with cytokine release syndrome (CRS) and immune effector cell-associated neurotoxicity syndrome [3].

Based on the above, CAR T-cell therapies for DLBCL were awarded innovation status by the Italian Medicines Agency, AIFA (Agenzia Italiana del Farmaco), and were automatically included on the regional formularies and funded through the national “Fondo per i Farmaci Innovativi Oncologici” (national fund for innovative oncology drugs), to provide equal access to eligible patients in Italy.

However, CAR-T introduction in clinical practice highlights the issue of economic sustainability, for both hospitals and the National Healthcare Systems—NHSs [12]. Moreover, despite the high investment in terms of therapy, a further element to be considered is the organizational impact within the hospitals that prescribe CAR-T.

The costs associated with these therapies are not limited to the acquisition costs alone. Other costs that will have a substantial impact on healthcare expenditures are hospitalization, intensive care unit (ICU) stays, as well as other costs related to the treatment of adverse events (AEs) and laboratory examinations. Furthermore, patients who live longer will also incur future medical costs unrelated to their conditions, for which they received CAR-T therapy. Conversely, longer survival and better disease control could facilitate pathology management and related family organization planning, as well as care-giver commitment, generating greater productivity levels and return-to-work activities [13,14].

Recent publications reported that public health experts, practitioners and clinicians worldwide called for Clinical Governance (CG) tools to evaluate CAR-T from an economic point of view, using cost-effectiveness criteria and economic forecast concerning the overall budget impact of CAR-T therapies [15,16,17].

Based on the above, an in-depth analysis of the economic and organizational sustainability of CAR-T therapies implementations is needed in the European setting, where the incidence rate of DLBCL is high and impactful, with a specific focus on Italy, where, on the one hand, spending review programs and cutting budgets were imperative for a long period [18], making the diffusion of technology-based healthcare innovations more difficult, and coupled with the COVID-19 pandemic that has severely impacted the National Healthcare Service, also including CAR-T delivery [19,20]. Thus, the present study aims at defining the economic and organizational impacts of CAR-T cells in Italy, for the management of DLBCL patients in third-line therapy, thus defining the level of sustainability for different healthcare stakeholders. An additional aim is the proposal and discussion of a potential reimbursement tariff.

In the attempt to achieve the above challenging objective, the authors would like to fill an important knowledge gap, regarding the provision of CAR-T therapy administration’s economic and organizational implications, that are not totally present in the recent literature and absolutely absent in the specific Italian context. Starting from the evaluation of the available literature evidence on the topic, an economic and organizational dimension analysis for the management of DLBCL patients was necessary. Indeed, the prior literature was focused exclusively on the cost of the CAR-T treatment at the NHS level, without an in-depth analysis (in economic terms) of the whole patient pathway, adding related implications and complications, also quantifying the CAR-T patient management costs directly impacting the hospitals.

This topic is then integrated with the proposal and discussion of a potential reimbursement tariff: this aspect is fundamental in the Italian NHS, which adopts a universalistic model, to ensure appropriateness and healthcare service quality.

In this regard, the estimation of costs could represent a relevant element, not only for an adequate healthcare resource allocation process, but also for supporting the production of proper reimbursement tariffs devoted to such patients requiring CAR T-cell administration, due to the current lack of a specific International Classification of Diseases (ICD) code to produce the appropriate Diagnosis-Related Group (DRG). This topic acquires a significant relevance since without any formal assessment concerning the financial aspects of these therapies, their costs remain intangible and vague.

Moving on from these premises, the aim of this study is to answer the following research question: *“Could CAR T-cell therapy represent a feasible and sustainable treatment option, both at the economic and organizational levels, to be offered to adult patients with relapsed or refractory DLBCL, assuming both the hospital and the NHS perspective?”*

## 2. Materials and Methods

In order to achieve the study objective, an analysis of the economic and organizational sustainability was developed since CAR-T therapy safety and efficacy profiles have been amply validated in the literature [9,10,11,21,22].

The analysis focused on the main treatments available for the management of DLBCL patients, comprising CAR-T cells and BSC, represented by a case-mix of treatment options used in the Italian clinical practice for the management of disease symptoms, also called Best Alternative Care, assuming the Italian hospital and NHS perspectives, over a 12-month time horizon.

Before starting the assessment of the economic and organizational dimensions, a narrative literature review was conducted to define the most important outcomes of safety and efficacy, which represent the driver of costs for the proper assessment of alternative technologies. As a first step of the evaluation of therapeutic alternatives, the safety and efficacy profiles were identified [21,22] following an approach widely accepted for multidimensional assessment. To achieve the study objective, some dimensions of the Health Technology Assessment (HTA) methodology were investigated, gathering real-world data, to evaluate processes and implications especially in economic and organizational terms. The HTA methodology is useful for decision makers, both at the institutional–macro level and the hospital–meso level, in the evaluation of health technologies [23].

The PICO (Problem/population, Intervention, Comparator and Outcome) approach was adopted [24].

(i)P (population): Patients affected by diffuse large B-cell lymphoma (DLBCL) in third-line therapy.(ii)I (intervention): CAR T-cell therapy.(iii)C (comparator): Chemotherapy—immunotherapy, Best Alternative Care, also defined as Best Salvage Care [5].(iv)O (outcome): Efficacy (OS and progression-free survival, PFS), safety (neurological adverse events and cytokine release syndrome), and related management costs.

The evidence in the literature came from a systematic search of databases (Cochrane Library, ClinicalTrials, PubMed, Prospero, and EMBASE,) considering the following keywords: “cancer”, “chemotherapy”, “oncologic*”, “adverse event”, “NH lymphoma”, “treatment”, “therapies”, “immunotherapy”, “blood”, “Tcells”, “remission”, “cells engineerization”, “ Best Salvage Care”, “Best Alternative Care”, “DLBCL”, “DLBCL relapse refractory”, “guidelines”, and “CAR-T”.

According to the PICO approach, peer-reviewed papers that explicitly described the clinical effectiveness and the safety profiles of the two treatments under assessment were included. Specifically, papers were consequently included and synthetized according to a PRISMA flow diagram [25] and their potential risk of bias and overall quality were assessed by means of the JADAD scale [26].

Once the safety and efficacy indicators were collected, the economic evaluation of DLBCL pathways was conducted by means of an activity-based costing approach [27], thus mapping all the activities conducted for the proper management of such a disease. The analysis considered the hospital point of view, in terms of the identification of the economic resource use directly sustained by hospitals in providing care for DLBCL patients.

For the definition of the overall economic resource absorption, a specific data extraction algorithm was developed (with reference to the year 2019). Only administrative data on services provided (including diagnostic and laboratory examinations, hospitalizations, outpatient procedures, and therapies) to 47 third-line patients with lymphoma were collected in two different Italian Hospitals.

All the above healthcare expenditure items derived from the anonymous administrative and accounting flows, provided by the management control of the hospitals involved, thus estimating DLBCL resource absorption, based on BSC or CAR T-cell administration. In addition, the clinical pathway derived from the above algorithm was then approved by a panel of nine experts composed of clinicians and pharmacists, adopting a Delphi approach [28], to ensure the scalability and generalizability of the pathway with respect to the different regional contexts.

For the economic assessment, the following hypotheses were made.

The cost of CAR-T therapy for the two drugs approved by the Italian Medicines Agency (AIFA) and currently used in the Italian market, for which AIFA have established “the payment by results mechanism” (i.e., the reimbursement of drug is related to the health results achieved), was referred to. For tisagenlecleucel, an initial payment equal to 30% is due at the infusion phase, whereas the remaining payment is due on the achievement of a successful and effective patient outcome (in particular, 35% at 6 months and 35% at 12 months). In contrast, a 50% reimbursement is due for axicabtagene ciloleucel at 6 months on achieving a successful and effective patient outcome. The remaining value is then reimbursed at 9 and 12 months, for 40% and 10% shares, respectively.The difference in the costs of CAR-T cells also emerged in the conservation phase. Tisagenlecleucel required a cryopreservation process, carried out by the hospital, while the collection and delivery costs related to the management of axicabtagene ciloleucel are supported by the manufacturer.CAR T-cell administration requires hospitalization of the patient. The economic evaluation considered the cost of hospitalization per day as EUR 1,875, and an overall length of stay equal to 15 days on average (analyzing gathered data, in line with literature evidence), for the management of patient monitoring and the infusion phase of the protocol for CAR-T treatment.BSC treatments, considered alternative therapeutic options, comprised the most common salvage therapy, excluding experimental protocols. In this specific setting, the most used therapeutic strategies are R-DHAX (rituximab, dexamethasone, cytarabine and oxaliplatin), R-GDP (rituximab, gemcitabine, dexamethasone, cisplatin, or carboplatin) and rituximab-bendamustine.Home palliative care and hospice care were included in the economic evaluation of BSC treatment.

The economic evaluation of the clinical pathway was integrated with the costs related to the management of the treatment-related adverse events, in terms of additional laboratory tests, diagnostic tests, drugs and hospitalizations, required to solve the patients’ complications. The adverse events occurrence rate was drawn from the literature evidence on the topic and the costs of treating such adverse events were based on the hospital-based data retrieved and validated by consensus, using a Delphi approach.

It should be noted here that the economic evaluation of DLBCL patients treated with CAR-T was stratified based on either being a responder or non-responder. In the latter, both home palliative care and hospice care costs were integrated in the economic evaluation. Furthermore, the assessment of CAR-T was properly carried out based on both the “payment by results” mechanism, thus adopting a financial approach, as well as without considering the treatment costs, and evaluating only the costs related to the care of the patients (thus considering the DLBCL patient management costs). In particular, the cost of the two main treatments—axicabtagene ciloleucel and tisagenlecleucel—was considered but, as above-mentioned, we applied the pay for results mechanism to achieve the correct final treatment value, approaching the hospital cost. Tisagenlecleucel was reimbursed at a rate of 30% for the first six months, followed by further reimbursement of 35% in the sixth month and 35% in the twelfth month, based on the effectiveness of the treatment for the patient. The average cost value was applied in the analysis, considering the survival rate based on evidence from the literature evidence. Axicabtagene ciloleucel, if effective, is reimbursed at a rate of 50% at the sixth-month time horizon, at 40% in the ninth month following treatment, and 10% in the twelfth month following the infusion. Furthermore, the final economic value depends on the survival rate of the patients. For the purposes of the present economic analysis, an average cost per patient of EUR 232,772.55 was applied. These data were directly gathered from the management control of the hospitals using the therapies, considering the efficacy of the CAR-T cells as presented in the literature evidence.

Once we defined the cost related to BSC and CAR-T administration, considering the aforementioned hypotheses, a budget impact analysis (BIA) was developed to define the impact of CAR-T implementation in clinical practice, thus supporting policy makers, in making long-term, system-wide, efficient decisions [29]. For the proper development of the BIA, a baseline scenario composed solely of BSC administration was compared to the introduction of CAR-T cells for patients presenting eligibility criteria, thus considering adult patients aged over 18 years old (Innovative Scenario 1) or over than 26 years old (Innovative Scenario 2), considering the two possible target populations, deriving from the indications for use of CAR-T. The final target population in the two scenarios strictly depends on the prevalence of the disease with respect to the age group and the median age of diagnosis, resulting in a different overall amount of potentially treatable populations.

An assumption in developing the budget impact analysis is that all the patients eligible for CAR-T in the Innovative Scenarios were treated with innovative therapy (this approach is defined as the complete replacement rate of the innovative technology or a 100% replacement rate).

Before starting the sustainability analysis, the target population eligible for CAR-T was defined based on the most recent epidemiological disease data. Indeed, DLBCL has an incidence of approximately 4.8 cases per 100,000 inhabitants, amounting to 2394.48 cases in Italy (starting from the population resident in Italy over 18 years—ISTAT, 2020). Following the Italian guidelines in terms of the rate of non-responders to the first line of therapy (30%), the rate of relapses within 2 years (10%), the rate of non-responders of refractory patients (82.5%) and relapsed patients (50%), the number of people eligible for the third line of treatment with CAR-T was equal to 219.69 patients in Italy (calculated as a sum). The target population eligible for CAR-T treatment was thus equal to 220 individuals aged over 18 years old and 202 individuals aged over 26 years old for the first year of treatment, while 215 individuals aged over 18 years old and 201 individuals aged over 26 years old for the second year of treatment, and finally 214 individuals aged over 18 years old and 200 individuals aged over 26 years old for the third year of treatment.

Furthermore, the BIA considered two scenarios, evaluating both the overall total costs, considering CAR T-cell therapy costs, and the cost of the DLBCL patient care pathway, excluding CAR T-cell therapy costs.

In conclusion, the organizational impact related to CAR-T introduction in the clinical practice was accordingly analyzed, with the assessment of the organizational investments required, in terms of additional staff, the learning curve, training courses and meetings, as well as in terms of new equipment or furniture purchases, based on a 12-month time horizon.

The assessment of the organizational investment was made through the administration of structured questionnaires, involving nine Italian healthcare professionals with specific technical knowledge in CAR-T administration and hematological diseases, covering different professional roles (clinicians and pharmacists). An anonymous structured questionnaire was administered through Lime Survey and based on the information of the professionals involved, both a minimum and a maximum organizational impact were calculated. This was useful because for under-discovered research areas, the collection of healthcare professionals’ perceptions attempts to fill in the gaps that are left unexposed by structured literature evidence [30,31,32].

## 3. Results

### 3.1. Results from the Literature Review

The search strategy identified 55 papers. Of these, only six were used in the present analysis [5,9,33,34,35,36], in accordance with the adopted strategy search approach (Figure 1).

The literature review showed a lack of scientific evidence concerning head-to-head comparisons between treatments. The implementation of the JADAD scale revealed that all the papers included for the definition of therapy safety and efficacy profiles were assessed as good quality and had a low risk of bias. All papers achieved a JADAD score equal to 4 (out of 5 maximum), thus the results are highly replicable in the proposed setting.

From an efficacy perspective (Table 1), the literature reported the incremental percentage in terms of both OS for CAR-T therapy (67%, [9]) vs. BSC (28%, [5], 2017), and in terms of progression-free survival (PFS) (66% [34]) vs. BSC (28%).

Table 2 and Table 3 report the adverse event incidence rates derived from the literature and the related economic evaluation (derived from the administrative data available and validated by the Delphi approach) for both CAR-T therapy and BSC.

### 3.2. Results from the Economic Evaluation

Table 4 reports the economic evaluation of the clinical pathways related to CAR-T and BSC. The results showed that the BSC clinical pathway required less resources in comparison to CAR-T (excluding the cost related to the therapy) (BSC: 29,558.41 vs. CAR-T: EUR 71,220.84, −58.5%).

In the stratification of CAR-T treatment based on being responders or non-responders, it emerged that the overall resources used for the treatment of a responder patient is equal to EUR 378,735.47, whereas the overall resources used for the treatment of a non-responder patient is equal to EUR 162,022.84. If the cost of CAR-T is not included, the economic analysis revealed a cost of EUR 71,220.84 for a responder patient and of EUR 74,902.84 for a non-responder patient, thus considering only the medical cost for the proper DLBCL patient’s management and care.

Once we defined the economic evaluation for patients of both BSC and CAR T-cell therapy, a BIA was performed to demonstrate the sustainability of the innovative therapeutic strategy.

Table 5 depicts that, at a national level, the introduction of CAR-T would generate an overall increase in costs, considering the CAR-T cost of the treatment (with a pay for performance reimbursement approach for the first year after the infusion, and introducing the level of efficacy, from the literature evidence, in the evaluation).

Table 6 depicts that, even excluding the direct cost related to CAR-T therapy, the introduction of the innovative care pathway would generate an overall increase in costs ranging from 15% to 23%, strictly dependent on the target population, and demonstrating the sustainability of the treatment, compared with the level of efficacy gained.

### 3.3. Results from the Organizational Assessment

The assessment of the organizational impact reveals that the introduction of the innovative CAR-T therapy would require additional investment, particularly related to the learning curve of the professionals involved in its administration. Notably, the sample agreed on the need to train all healthcare professionals on introducing the new technologie, as well as to hold hospital meetings to determine the proper patient clinical pathway. Indeed, staff with different backgrounds (on average, 8 clinicians, 2 pharmacists and 12 nurses working in a hematological department) are required to attend training courses and hospital meetings for one day, on average.

Part of the professional sample declared the need for further structural investment to ensure the presence of additional ICU beds devoted for patients specifically treated with CAR-T therapy: this could be considered the case for a medium–large-sized hospital, with hub center activities, needing to ensure, in the long run, the proper management of all the phases of CAR-T treatment introduction.

Based on the above consideration, the implementation of CAR-T treatments would require some additional investments, equal to a minimum of EUR 15,868.63 for a medium–large-sized hospital, with all the requirements imposed for accreditation by the Joint Commission International (JCI). This investment would increase, becoming equal to EUR 102,949.49, on the introduction of an additional hospital bed in an ICU, as reported in Table 7, suggesting a dedicated ICU bed for the management of the more severe adverse events or in case of a high volume of CAR-T patient management activities.

The variability of such organizational investments is due to the fact that the minimum scenario considered the necessary and sufficient conditions to be able to manage DLBCL patients being administered CAR-T therapy, thus revealing the capability of the hospital to respond to the daily management of such patients, and considering a 12-month time horizon after CAR-T introduction in the clinical practice.

On the other hand, the maximum scenario of organizational investment would also include the structural investment required in the medium- and long-term period, especially in regional or hub hospitals.

## 4. Discussion

The treatment of refractory DLBCL patients has been an active area in the hematologic research field, given the dismal prognosis and the poor outcomes. Thus, any treatment strategy able to achieve an outcome improvement acquires strategic relevance for DLBCL patient pathways. The cost burden needs to be evaluated to ensure the sustainability of the public healthcare policies in this field.

Total cost estimations are wide ranging and depend in large part on treatment modality. Standard first-line care alone incurs considerable expenses. Patients with relapsed/refractory disease will require additional costly therapies including CAR-T cells, leading to dramatic increases in per-patient fees and financial strain on hospitals. Exciting advances in DLBCL treatment and surveillance have the potential to improve outcomes, but these technological advances may yield greater costs as well.

DLBCL prevalence is likely to increase with the ageing population, and determination of cost-effective first-line and later-line therapies and surveillance modalities in this pathology will require continued economic evaluation to limit the significant financial burden placed on patients and hospitals in the treatment of DLBCL.

The results of the present study show new economic evidence for healthcare decision makers, at the hospital, regional and national levels, to optimize the appropriateness of resource allocation, thus reporting findings from an Italian experience.

From an economic perspective, the evaluation of CAR-T is consistent with literature evidence, taking into consideration the setting of the EU-5 (France, Germany, Spain, Italy, and the United Kingdom) and the Netherlands [15,16,17]. A difference emerged concerning the costs related to pre- and post-treatments equal to EUR 50,359 per patient (vs. the value of EUR 70,859.85 calculated in the present study): this difference may be explained due to the integration of the impact of adverse events and to the related economic evaluation. Other variations concern the economic value devoted to CAR T-cell therapy (ranging from EUR 307,200 to EUR 350,000, vs. EUR 232,772.55 used in the present study, where the “payment by results” mechanism was applied), since the economic value is often related to the local adaptation of data retrieval.

The specific focus on CAR-T resource absorption suggests the need to introduce a reimbursement tariff that is dedicated and adequate, both at the hospital and NHS levels, for the new CAR-T pathway. Indeed, CAR-T therapy could be reimbursed through the “payment by results” mechanism: this method could support the Italian NHS in the effective use of the specific budget for “innovative drugs” where there is demonstrated therapeutic effectiveness over time.

Instead, the remuneration model for the patient pathway should be redesigned to ensure that the adverse events do not excessively burden, especially in the initial phase, the introduction of CAR-T treatment in hospitals.

This is particularly relevant since no consensus exists, at least in the Italian setting, concerning the proper reimbursement tariff devoted to such procedures. The most used DRG is represented by the Italian DRG n. 481 “Bone marrow transplant”, which presents a reimbursement tariff equal to EUR 59,806 (Italian Ministry of Health, Decree 18 October 2012, Remuneration for acute hospital care, rehabilitation and post-acute hospital care and specialist outpatient care), and this is is not consistent with all the activities required for the proper implementation of CAR-T and the consequent overall management of eligible individuals. This consideration would suggest the remodulation of the reimbursement tariff in the Italian setting, as in the Lombardy Region (Lombardy Region Decree no. VII/941 3 August 2000 (Update of the tariffs of hospital services, in hospitalization), which stratifies the DRG n. 481 tariff based on the clinical condition of the patients: (i) Type “A”—intensified chemotherapy with support of autologous peripheral stem cells: when the procedure code ICD-9CM 41.04 is used in the absence of myeloablative conditioning; (ii) Type “B”—autologous stem cell transplantation after myeloablative conditioning: when the procedure code ICD-9CM 41.01 or the code 41.04 is used after myeloablative conditioning; (iii) Type “C”—HLA-compatible allogeneic bone marrow transplant from consanguineous: when the procedure code ICD-9CM 41.02 or the code 41.03 is used; and (iv) Type “D”—incompatible inbred allogeneic bone marrow transplant and unrelated allogeneic bone marrow transplant, including umbilical cord transplant: when the procedure code ICD-9CM 41.02 or the code 41.03 is used.

In the Delphi approach, experts pointed out that a specific pathway for patients with CAR-T who are discharged from the hospital is necessary: patients could present adverse events that could be managed in the outpatient setting to avoid infections and optimize the organizational capacity of Hospital Departments involved in patient care.

Moving on from these premises, the results revealed that the reimbursement strategy of CAR-T should be redesigned from an economic-organizational point of view to remunerate the hospitals who are capable of guaranteeing this innovative pathway, assuming significant risks related to the need for timely management of adverse events.

This consideration is also confirmed by Kron and colleagues [16], who have tried to analyze the economic, procedural, and organizational CAR T-cell pathway, using specific tools to evaluate the reimbursement and remuneration of the procedure in Germany, taking into consideration the service provider point of view. Results from the cited study revealed that CAR T treatment received insufficient reimbursement, due to the high drug costs as well as the necessary investment in personnel and infrastructure.

Despite the relevance of the topic, the present research had some limitations that should be mentioned. The generalizability and the transferability of the results to other settings and thus the conclusions of this study should be considered with caution, because data collection was performed in a geographically focused setting (Italy). For example, the limitations of the Italian reimbursement are mainly related to the fact that the negotiated prices are confidential and that not all markets use the payment by results approach.

Moreover, the economic evaluation of the CAR-T pathway was mainly based on expert opinion, due to the limited availability of clinical data, as was also conducted in other studies [17,18,19,20,21,22,23,24,25,26,27,28,29,30,31,32,33,34,35,36,37,38]. In addition, the economic burden considered for CAR-T therapies is based on a “payment by result” approach specifically adopted in the Italian context. This situation could be different in other contexts, and therefore the costs, if paid up front at the time of treatment, could be higher for non-responder patients in comparison to those proposed in the present analysis.

However, the results reported economic patterns based on real-world evidence and practices in Italy, with important implications for decision makers.

As also confirmed by the literature, CAR T-cell therapies are clinically effective, being a promising treatment option for DLBCL, but the financial burden on healthcare systems is high and expenditure will rise in the hematology setting. Thus, to narrow this burden, the development of interactions and collaborations between hospital decision makers and pharmaceutical manufacturers could be a good strategy.

## 5. Conclusions

The results of this study generate an interesting economic and organizational contribution, thus covering an important knowledge gap, concerning both the definition of the real-life economic impact of CAR T-cell therapy, as a treatment option to be offered to DLBCL patients, and the never-ending divide between costs sustained and reimbursement tariffs proposed and settled by the healthcare system. The main strength of the present study is that it provides a “real-life” picture of the potential implications of CAR-T in Italian clinical practice, also offering an overview of the consequent economic resource use and reimbursement tariff estimation of the patient clinical pathway, completed and enriched using robust evidence, a stream of literature and an active discussion in the Italian policy makers’ agendas [39,40]. To the best of the authors’ knowledge, this study has something new to add to the scientific literature on the topic.

Indeed, compared with the available and recent literature, the research contribution of this paper consists of: (i) showing new economic and organizational evidence for healthcare decision makers, at the hospital, regional and national levels, to optimize the appropriateness of resource allocation; (ii) using rigorous methodologies, approved and largely used in the scientific literature for the findings; (iii) taking a holistic perspective of the investigated phenomenon and not considering only the cost of acquisition of the two therapeutic options but the entire clinical pathway and the related treatment-adverse events; (iv) proposing a reimbursement strategy for the CAR-T option.

In this view, the first contribution of the work, compared with the existing literature, consists of an accurate and in-depth analysis of the economic and organizational impact of the treatment of refractory DLBCL patients with different therapeutic options. These data and information could be useful to decision makers in programming and managing the available resources.

The second contribution consists of the use of rigorous methodologies, such as process mapping, economic evaluation of the whole clinical pathway (taking into account an activity-based costing approach [27]), budget impact analysis [29], and the analysis of investment at the hospital level devoted to the introduction of the innovative CAR-T therapy, quantified as the organizational impact. The mentioned methodologies are normally proposed in the implementation of Health Technology Assessment [23] studies. The implementation of this study design approach could be judged as an aspect of innovation, in comparison to the available literature evidence [15,16,17,41,42,43]. In particular, the HTA approach combines the collection of evidence-based information and original data.

The third contribution is the adopted perspective: the current literature [15,41,42,43], mainly focuses only on the cost-effectiveness or cost-utility of the innovative treatment option (CAR-T cells), without taking into consideration a holistic perspective, unlike our work that examined the entire patient clinical pathway and the treatment-related adverse events of the two treatment options (CAR-T cells and BSC). From an economic perspective, we tried to define the overall economic resource absorption devoted to the management of DLBCL patients treated with CAR-T or best supportive care. We are aware that past studies have evaluated the economic impact of such innovative therapies, but the evidence focused only on the costs related to CAR-T administration, without integrating the monitoring activities, in terms of follow-up procedures performed with patients after administration, or the management of adverse events occurring after treatment. First of all, this analysis convers a growing concern in the Healthcare System—the burden of expenses related to CAR-T therapies. In addition, based on the fact that the rates of occurrence of adverse events derived from literature evidence and were not collected within Italian clinical practice, this economic information could represent the baseline cost for a DLBCL patient receiving CAR-T therapy, supporting hospital benchmarking activities.

The costs for the treatment-related adverse event management considered additional laboratory tests, diagnostic tests, additional treatments and hospitalizations. This information is relevant, informing and giving decision makers the possibility to also evaluate the resources required to solve patient complications. These therapies are not limited to the acquisition costs alone or the definition of the cost-effectiveness, as already studied in the scientific evidence. Indeed, studying treatment-related adverse event management, other costs resulted in a substantial impact on healthcare expenditure, such as hospitalizations and intensive care unit (ICU) stays. Moreover, it is important to remember that patients who live longer will also incur future medical costs unrelated to their conditions, for which they receive CAR-T therapy.

The fourth contribution of our paper is the proposal of the reimbursement strategy of CAR-T treatment redesign: the scientific literature has not yet explored this important topic. In particular, the presented findings could stimulate a debate about the definition of a proper reimbursement tariff. The proper reimbursement tariff is that which may cover the entire costs directly sustained by hospitals. The approach used in the manuscript could be valid not only in Italy, but also in other countries: in this view, the present findings could also be useful for decision makers of other NHS similar to that in Italy.

Another contribution related to the first is from an organizational aspect, because no evidence exists to date on the quantification of CAR-T patient management costs directly impacting hospitals, thus focusing on the definition of CAR-T organizational sustainability, in terms of investments required to deliver this treatment. In fact, CAR T-cell therapies must be provided by centers that fulfil the minimal requirements authorized by AIFA, namely “Certification by the National Transplant Center” in line with EU directives. JACIE accreditation for allogenic transplantation includes clinical, cell collection and cell processing units; availability of an intensive care and reanimation unit; and presence of a multi-disciplinary team adequate for management of the therapy and possible adverse events. In addition to organizational and training needs, significant investment in center infrastructure is often necessary—for instance, to expand intensive care unit (ICU) space or to adjust to specific CAR T-cell therapy logistical requirements.

Based on this consideration, our contribution is to provide an estimation of the organizational investment devoted to CAR-T hospital administration. As such, our analysis reported that the implementation of CAR-T treatments would require some additional investments, equal to a minimum of EUR 15,868.63 for a medium–large-sized hospital, with all the requirements imposed for the accreditation by the Joint Commission International (JCI). This investment would increase, becoming equal to EUR 102,949.49, on the introduction of an additional hospital bed in an ICU, suggesting a dedicated ICU bed for the management of more severe adverse events or in the case of a high volume of CAR-T patient management activities.

This aspect is also relevant to support the decision of a region to deliver CAR T-cell therapy, by the accreditation of a specific hospital. In this view, the region should decide where to invest in the treatment delivery or to send the patient requiring therapy to another region. This topic is acquiring strategic relevance since the Italian NHS lacks specific indications from regional and national authorities for necessary infrastructural and organizational investments which can leave CAR T centers carrying investment costs by themselves.

In conclusion, the presented topics (in particular, the economic and organizational aspects of CAR-T) are acquiring strategic relevance considering the recent National Recovery and Resilience Plan (NRRP), where public health intervention is among the major components. In particular, the NRRP would develop specific public health interventions able to enhance skills and human capital, as well as investment for digital, structural, and technological resources, thus promoting the renewal and modernization of the existing technological and digital healthcare structures. In addition, the NRRP focuses its attention on allocating resources for the digital transition, the employment of innovative health technologies, and the strengthening of intensive and semi-intensive care units, thus also being useful for proper management of DLBCL patients eligible for CAR T-cell therapy.

## Figures and Tables

**Figure 1 ijerph-20-03830-f001:**
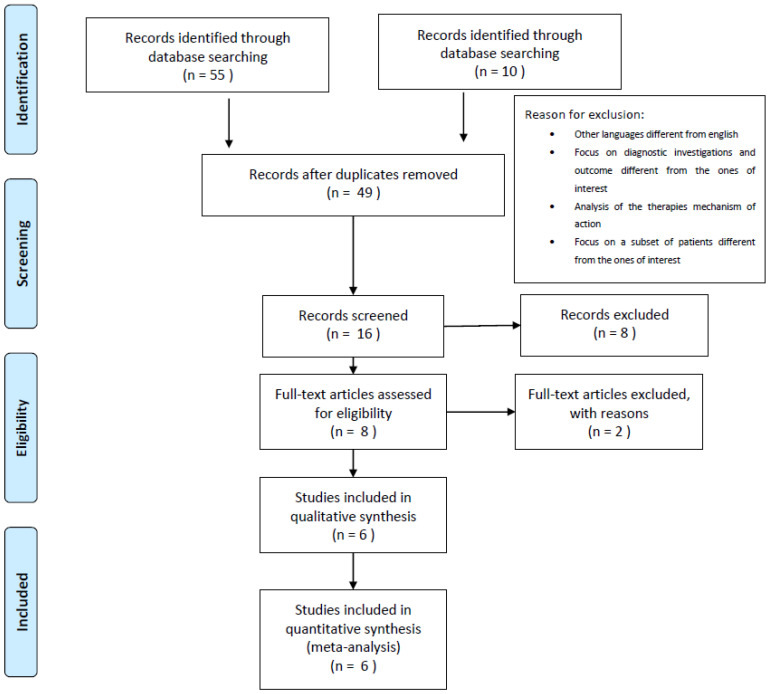
Prisma flowchart.

**Table 1 ijerph-20-03830-t001:** Efficacy parameters: overall survival and progression-free survival at 12-months: CAR-T vs. BSC.

	CAR-T			BSC		
	Chavez et al. (2019) [34]	Locke et al. (2017) [33]	Neelapu et al. (2017) [9]	Barton et al. (2014) [36]	Crump et al. (2017) [5]	Arcari et al. (2016) [35]
**Overall Survival (12 months)**	49%	59%	67%	50%	28%	10.8 months
**Progression-Free Survival (12 months)**	66%	44%	42%	28%	-	8.8 months

**Table 2 ijerph-20-03830-t002:** CAR T-cell adverse event incidence rates reported in the literature and related economic evaluations.

	Adverse Event Incidence Rates	Reference	Economic Evaluation of the Adverse Events [EUR]
*Cytokine Release Syndrome*			
Pyrexia	86%	Locke et al., 2017 [33]	EUR 232.97
Hypotension	71%	Locke et al., 2017 [33]	EUR 963.00
Tachycardia	43%	Locke et al., 2017 [33]	EUR 124.59
Acute kidney injury	29%	Locke et al., 2017 [33]	EUR 5167.40
Cardiac failure	14%	Locke et al., 2017 [33]	EUR 4700.00
Metabolic acidosis	14%	Locke et al., 2017 [33]	EUR 774.97
Hyponatremia	14%	Locke et al., 2017 [33]	EUR 774.97
Dyspnea	21%	Locke et al., 2017 [33]	EUR 1484.00
Infection	14%	Locke et al., 2017 [33]	EUR 447.97
*Immune effector cell-associated neurotoxicity syndrome*			
Headache	14%	Locke et al., 2017 [33]	EUR 20.07
Hypoxia	14%	Locke et al., 2017 [33]	EUR 709.38
Encephalopathy	37%	Locke et al., 2017 [33]	EUR 2077.00
Tremor	31%	Locke et al., 2017 [33]	EUR 709.38
Agitation	14%	Locke et al., 2017 [33]	EUR 709.38
Aphasia	14%	Locke et al., 2017 [33]	EUR 709.38
Delirium	14%	Locke et al., 2017 [33]	EUR 709.38
Dizziness	14%	Locke et al., 2017 [33]	EUR 113.31
Hallucination	14%	Locke et al., 2017 [33]	EUR 709.38
Restlessness	14%	Locke et al., 2017 [33]	EUR 709.38
Nausea	58%	Locke et al., 2017 [33]	EUR 296.28
Vomiting	34%	Locke et al., 2017 [33]	EUR 52.74
Constipation	30%	Locke et al., 2017 [33]	EUR 153.96
Confused state	27%	Locke et al., 2017 [33]	EUR 709.38
*Haematological events*			
Anemia	68%	Neelapu et al., 2017 [9]	EUR 23,625.51
Thrombocytopenia	35%	Neelapu et al., 2017 [9]	
Decreased neutrophil count	33%	Locke et al., 2017 [33]	
Decreased platelet count	30%	Locke et al., 2017 [33]	
Increased alanine aminotransferase	20%	Locke et al., 2017 [33]	
Decreased lymphocyte count	20%	Locke et al., 2017 [33]	
Leucopenia	19%	Locke et al., 2017 [33]	

**Table 3 ijerph-20-03830-t003:** BSC adverse event incidence rates in the literature and related economic evaluation.

	Adverse Event Incidence Rates	Reference	Economic Evaluation of the Adverse Events [EUR]
Neutropenia	60% 64%	Arcari et al., 2016 [35] Barton et al., 2015 [36]	EUR 1678.90
Anemia	45%	Arcari et al., 2016 [35]	EUR 7320.93
Thrombocytopenia	29% 69%	Arcari et al., 2016 [35] Barton et al., 2015 [36]	EUR 1349.00
Infection	27%	Arcari et al., 2016 [35]	EUR 447.97
Nausea	25%	Arcari et al., 2016 [35]	EUR 106.74
Anorexia	18%	Arcari et al., 2016 [35]	EUR 22.50
Fatigue	5%	Arcari et al., 2016 [35]	EUR 22.50
Diarrhea	4%	Arcari et al., 2016 [35]	EUR 153.96
Maculopapular rash	9%	Arcari et al., 2016 [35]	EUR 132.12
Dehydration	4%	Arcari et al., 2016 [35]	EUR 22.50
Squamous cell carcinoma of the skin	4%	Arcari et al., 2016 [35]	EUR 204.33

**Table 4 ijerph-20-03830-t004:** CAR-T and BSC treatment: process mapping and economic evaluation of the clinical pathways.

Process Mapping and Economic Evaluation of CAR-T Treatment	Mean Cost per Patient [EUR]
Procedures and controls cryo-conservation	361.00 EUR
CAR-T therapy	232,772.55 EUR
PET (Positron-Emission Tomography)	1081.86 EUR
Lymphodepleting chemotherapy	650.58 EUR
Lymphocyte collection + CAR-T infusion + observation in hospital	27,185.91 EUR
Neurotoxicity and cytokine release toxicity and Tocilizumab use	41,580.50 EUR
**Total with CAR-T treatment**	**303,632.40 EUR**
**Total without CAR-T treatment costs**	**71,220.84 EUR**
**Process Mapping and Economic Evaluation of BSC Treatment**	**Mean Cost per Patient [EUR]**
PET (Positron-Emission Tomography)	EUR 1081.86
Lymphodepleting chemotherapy	EUR 1971.26
Salvage chemotherapy	EUR 17,500.00
Adverse events	EUR 5176.28
Hospitalization in hospice and home palliative care (on average 20 days)	EUR 3682.00
**Total BSC treatment**	**EUR 29,558.41**

**Table 5 ijerph-20-03830-t005:** Budget impact analysis at the national (Italian) level considering the CAR-T cost of the treatment (with a pay for performance reimbursement approach for the first year after the infusion).

**I year**	**Scenario**	**Overall Costs for the Italian Cohort**	**Differences**
	Baseline Scenario	12,453,917.20 EUR	
	Innovative Scenario 1 (individuals over 18 years old)	53,769,548.98 EUR	332%
	Innovative Scenario 2 (individuals over 26 years old)	49,350,521.33 EUR	296%
**II year**	**Scenario**	**Overall Costs for the Italian Cohort**	**Differences**
	Baseline Scenario	12,310,871.42 EUR	
	Innovative Scenario 1 (individuals over 18 years old)	52,561,627.69 EUR	327%
	Innovative Scenario 2 (individuals over 26 years old)	49,374,005.83 EUR	301%
**III year**	**Scenario**	**Overall Costs for the Italian Cohort**	**Differences**
	Baseline Scenario	12,253,833.80 EUR	
	Innovative Scenario 1 (individuals over 18 years old)	52,318,055.43 EUR	327%
	Innovative Scenario 2 (individuals over 26 years old)	49,145,289.79 EUR	301%
**Total**	**Scenario**	**Overall Costs for the Italian Cohort**	**Differences**
	Baseline Scenario	37,018,621.41 EUR	
	Innovative Scenario 1 (individuals over 18 years old)	158,649,230.60 EUR	329%
	Innovative Scenario 2 (individuals over 26 years old)	147,869,815.38 EUR	299%

**Table 6 ijerph-20-03830-t006:** Budget impact analysis at national (Italian) level not considering the CAR-T cost of the treatment.

**I year**	**Scenario**	**Overall Costs** **for the Italian Cohort**	**Differences**
	Baseline Scenario	12,453,917.20 EUR	
	Innovative Scenario 1 (individuals over 18 years old)	15,430,415.83 EUR	24%
	Innovative Scenario 2 (individuals over 26 years old)	14,162,273.97 EUR	14%
**II year**	**Scenario**	**Overall Costs for the Italian Cohort**	**Differences**
	Baseline Scenario	12,310,871.42 EUR	
	Innovative Scenario 1 (individuals over 18 years old)	15,083,774.89 EUR	23%
	Innovative Scenario 2 (individuals over 26 years old)	14,169,012.11 EUR	15%
**III year**	**Scenario**	**Overall Costs for the Italian Cohort**	**Differences**
	Baseline Scenario	12,253,833.80 EUR	
	Innovative Scenario 1 (individuals over 18 years old)	15,013,875.49 EUR	23%
	Innovative Scenario 2 (individuals over 26 years old)	14,103,377.75 EUR	15%
**Total**	**Scenario**	**Overall costs for the Italian cohort**	**Differences**
	Baseline Scenario	37,018,621.41 EUR	
	Innovative Scenario 1 (individuals over 18 years old)	45,528,064.62 EUR	23%
	Innovative Scenario 2 (individuals over 26 years old)	42,434,662.82 EUR	15%

**Table 7 ijerph-20-03830-t007:** Definition of the organizational investment.

Organizational Investment	Unit Number [Minimum]	Unit Number [Maximum]	Typology of Investment	h Min	h Max	Euro/h	Euro/unit	Organizational Investment [Minimum]	Organizational Investment [Maximum]
**Additional Staff**	0	1	Nurse				EUR 39,398.40	EUR -	EUR 39,398.40
**Training Course**	20	25	Clinician	6	10	EUR 39.60		EUR 4752.00	EUR 9900.00
	20	25	Nurse	6	10	EUR 21.60		EUR 2592.00	EUR 5400.00
	2	2	Pharmacist	6	10	EUR 34.20		EUR 410.40	EUR 410.40
	2	4	Laboratory Experts	6	10	EUR 21.60		EUR 259.20	EUR 864.00
	2	4	Biologist	6	10	EUR 21.60		EUR 259.20	EUR 864.00
**Hospital Meeting**	20	25	Clinician	4	24	EUR 39.60		EUR 3168.00	EUR 23,760.00
	20	25	Nurse	4	24	EUR 21.60		EUR 1728.00	EUR 12,960.00
	2	2	Pharmacist	4	24	EUR 34.20		EUR 273.60	EUR 1641.6
	2	4	Laboratory Experts	2	4	EUR 21.60		EUR 86.40	EUR 345.60
	2	4	Biologist	2	4	EUR 21.60		EUR 86.40	EUR 345.60
**Patient and Caregiver education**	3	4	Clinician	1	2	EUR 39.60		EUR 118.80	EUR 316.80
	2	4	Nurses	1	2	EUR 21.60		EUR 43.20	EUR 172.80
**Additional Furniture**	0	1	Hospital Beds				EUR 296.00	EUR -	EUR 296.00
**Additional Equipment**	1	3	Multi-parameter patient monitors				EUR 2091.43	EUR 2091.43	EUR 6274.29
**Total costs**								**EUR 15,868.63**	**EUR 102,949.49**

## Data Availability

Data are available on request from the authors.

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
