# Peer review of "Multidimensional Results and Reflections on CAR-T: The Italian Evidence"

_ijerph, 2023, doi:10.3390/ijerph20053830_

Round 1
Reviewer 1 Report (Previous Reviewer 1)
After carefully reading the resubmited manuscript, I still find some critical problems have not been well addressed.
(1) The research gap this manuscript would like to fill is unclear, and readers know little about "what research findings prior investigations have achived?", "what's the limitation prior studies have left?", "what's the research contribution current study has made?"
(2) The current study seems like an introductory review of six mate-analyses (figure 1). The reserach contribution is limited.
Author Response
Replies to Reviewer #1
We thank Reviewer # 1 for all the comments and interesting suggestions received. We addressed them, as detailed below.
Comments and Suggestions for Authors
(1) The research gap this manuscript would like to fill is unclear, and readers know little about "what research findings prior investigations have achived?", "what's the limitation prior studies have left?", "what's the research contribution current study has made?"
We would like to thank the Reviewer # 1 for the kind comment and for the possibility to improve this aspect in the manuscript.
The gap that authors would like to fill with the present manuscript is to provide the economic and organizational implications related to CAR-T therapy administrations, that are not totally present in the recent literature and absolutely absent for the specific Italian context. Starting from the analysis of the literature evidence available on the topic, the development of the economic and organizational dimensions analysis for the management of DLBCL patients was necessary to fill this knowledge gap.
In fact, prior literature was focused exclusively on the cost of the CAR-T treatment at NHS level, without a deepened analysis (in economic terms) of the whole patient pathway, in terms of overall economic resources absorption for the management of patients eligible to CAR-T therapy, thus integrating all the monitoring activities and the resolution of potential adverse events and complications that may occur after treatment, assuming a 12-month time horizon. Furthermore, no evidence exists up to date with regard to the quantification of CAR-T patient management costs directly impacting on the hospitals, thus focusing on the definition of CAR-T organizational sustainability.
Another gap we would like to fill is related to the proposal and discussion of a potential reimbursement tariff: this aspect is fundamental in the Italian NHS, that adopts a universalistic model, to ensure the appropriateness and healthcare services quality.
We added these considerations in the Introduction section of the amended manuscript.
(2) The current study seems like an introductory review of six mate-analyses (figure 1). The reserach contribution is limited.
We would like to thank the Reviewer # 1 for the kind comment and for the possibility to improve this aspect in our manuscript.
The research contribution consists in showing new economic and organizational evidence for healthcare decision-makers, at hospital, regional and national level, to optimize the appropriateness of resources allocation. We report findings from an Italian experience of the treatment of refractory DLBCL patients.
New findings that have been achieved in our manuscript consist of an in-depth economic analysis related to the Italian setting (i.e. process mapping and economic evaluation of the considered clinical pathways, Budget Impact Analysis, the investment at hospital level devoted to the introduction of the innovative CAR-T therapy, quantified as organizational impact), without focusing the attention only on the definition of the cost-effectiveness or cost-utility nature of such innovative treatment options. For the economic evaluation of the patient clinical pathways, another contribution of the present manuscript is related to the possibility to evaluate and inform on the costs for the treatment-related adverse events management, in terms of additional laboratory tests, diagnostic tests, additional treatments and hospitalizations, required to solve the patients’ complications.
In fact, as previously mentions, scientific evidence available on the topic focused the attention of the identification of CAR-T cell acquisition costs or on the definition of the cost-effectiveness nature of such innovative therapy (Heine et al., 2021; Borgert et al., 2021; Fiorenza et al., 2020 Lin et al., 2019). However, the costs associated with these therapies are not limited to the acquisition costs alone. Other costs may have a substantial impact on healthcare expenditures are hospitalization, Intensive Care Unit (ICU) stays, as well as other costs related to the treatment of adverse events (AEs) and laboratory examinations. Furthermore, patients who live longer, will also incur future medical costs unrelated to their conditions, for which they received CAR-T therapy.
In addition to the definition of the overall management cost related to DLBCL patients eligible to CAR-T cell therapy, assuming both the hospital and the NHS perspective, another contribution of our paper is the proposal of the redesign of the reimbursement strategy of CAR-T, thus being a topic that has not been explored yet. In particular, the results would stimulate a debate with concerning the definition of a proper reimbursement tariff that may cover the costs directly sustained by hospitals, since the economic results here presented derived from data collection conducted in real world settings, useful also for decision makers of other Universal NHS as Italy.
This approach enables able to combine the collection of evidence-based information and original data, for a specific local adaptation of the evidence and findings, in particular, for economic and organizational results, and it is normally proposed in the implementation of Health Technology Assessment study. The implementation of this study design approach represents an aspect of innovation, in comparison to the available literature evidence (Heine et al., 2021; Kron et al, 2021; Raimond et al., 2021; Borgert et al., 2021; Fiorenza et al., 2020 Lin et al., 2019.
We added these considerations in the Conclusion section of the revised manuscript.

Round 2
Reviewer 1 Report (Previous Reviewer 1)
The statement of research contribution remains unclear and difficult to follow. The present research analysis is difficult to achieve the proposed research goal or contribution. The present research results mainly come from prior research and the original contribution seems marginal.
Author Response
Replies to Reviewer #1
We thank Reviewer # 1 for the comment received. We addressed it, as detailed below.
Comments and Suggestions for Authors
The statement of research contribution remains unclear and difficult to follow. The present research analysis is difficult to achieve the proposed research goal or contribution. The present research results mainly come from prior research and the original contribution seems marginal
We would like to thank the Reviewer # 1 for the kind comment and for the possibility to improve this aspect in our manuscript that could be unclear for an external reader.
We tried to explain better this issue in relationship to our findings, taking into consideration real data and information related to the Italian experience of the treatment of refractory DLBCL patients with different options (Best Salvage Care-BSC and CART-cells).
We used literature evidence in our study, but we strongly believe to have add also new evidence to the scientific literature about the topic.
Indeed, compared with the available scientific literature, in our opinion, the research contribution of the paper consists in: i) showing new economic and organizational evidence for healthcare decision-makers, at hospital, regional and national level, to optimize the appropriateness of resources allocation; ii) using rigorous methodologies, approved and largely used in scientific literature, to achieved the findings; iii) taking an holistic perspective of the investigated phenomenon and not considering only the cost of acquisition of the two therapeutic options but the entire clinical pathway and the related treatment-adverse events; iv) proposing a reimbursement strategy of CAR-T option.
Please note that the economic and organizational findings presented in the manuscript derived from an in-depth analysis of the Italian context and regulation.
In this view, the first contribution of the work, compared with the existent literature, consists in an accurate and in-depth analysis of the economic and organizational impact of the treatment of refractory DLBCL patients with different therapeutic options. These data and information could be useful to decision makers in programming and manage the available resource.
The second contribution consists in the use of rigorous methodologies, such as process mapping, economic evaluation of the whole clinical pathways (taking into account an Activity Based Costing approach (Cooper and Kaplan, 1992)), Budget Impact Analysis (Sullivan et al., 2014), the analysis of investment at hospital level devoted to the introduction of the innovative CAR-T therapy, quantified as organizational impact. The mentioned methodologies are normally proposed in the implementation of Health Technology Assessment (EUNetHTA) studies. The implementation of this study design approach could be judged as an aspect of innovation, in comparison to the available literature evidence (Heine et al., 2021; Kron et al, 2021; Raimond et al., 2021; Borgert et al., 2021; Fiorenza et al., 2020 Lin et al., 2019). In particular, HTA approach combine the collection of evidence-based information and original data.
The third contribution is the adopted perspective: the current literature (Heine et al., 2021; Borgert et al., 2021; Fiorenza et al., 2020 Lin et al., 2019), mainly focuses the attention only on the cost-effectiveness or cost-utility of the innovative treatment option (CAR-T cells), without taking into consideration a holistic perspective, like our work that examined the entire patient clinical pathway and the treatment-related adverse events of the two treatment options (CAR-T cells and BSC).
From an economic perspective, we tried to define the overall economic resources absorption devoted to the management of a DLBCL patients, treated with CAR-T or best supportive care. We are aware that past studies evaluate the economic impact of such innovative therapies, but the evidence focused the attention only on the cost related to CAR-T administration, without integrating the monitoring activities, in terms of follow-up procedure performed to patients after the administration, or the management of adverse events occurring after treatment. First of all, this analysis convers a growing concern on Healthcare System, that is the burden of expenses related to CAR-T therapies. In addition, based on the fact that the adverse events occurrence rates derived from literature evidence and were not collected within the Italian clinical practice, this economic information could represent the baseline cost for a DLBCL patient receiving CAR-T therapy, supporting hospitals’ benchmarking activities.
The costs for the treatment-related adverse events management observed additional laboratory tests, diagnostic tests, additional treatments and hospitalizations. This information is relevant, giving possibility to evaluate and inform decision makers also about the resources required to solve the patients’ complications. The costs associated with these therapies are not limited to the acquisition costs alone or the definition of the cost-effectiveness, as already study in scientific evidence. Indeed, studying the treatment-related adverse events management, we found that other costs may have a substantial impact on healthcare expenditures like hospitalization, Intensive Care Unit (ICU) stays. Moreover, it is important to remember that patients who live longer, will also incur future medical costs unrelated to their conditions, for which they received CAR-T therapy.
The fourth contribution of our paper is the proposal of the redesign of the reimbursement strategy of CAR-T treatment: the scientific literature has not been explored yet this important topic. In particular, our findings could stimulate a debate about the definition of a proper reimbursement tariff. The proper reimbursement tariff, indeed, is that may cover the entire costs directly sustained by hospitals. The approach used in the manuscript could be valid not only in Italy, but it could be generalized also in other country: in this view, the present findings could be useful also for decision makers of other Universal NHS as Italy.
In relationship to the first contribution the organizational analysis is notable because no evidence exists up to date with regard to the quantification of CAR-T patient management costs directly impacting on the hospitals, thus focusing on the definition of CAR-T organizational sustainability, in terms of investments required to deliver this treatment. In fact, CAR T-cell therapies must be provided by centers authorized by the regions, which fulfil the minimal requirements from AIFA, namely: “Certification by the National Transplant Center” in line with EU directives. JACIE accreditation for allogenic transplantation including clinical, cell collection and cell processing units; availability of an intensive care and reanimation unit; and presence of a multi-disciplinary team adequate for management of the therapy and possible adverse events. In addition to organizational and training needs, significant investment in center infrastructure is often necessary, for instance to expand intensive care unit (ICU) space or to adjust to specific CAR T-cell therapy logistical requirements.
Based on this consideration, the contribution is to provide an estimation of the organizational investment devoted to CAR-T hospital administration. Our analysis reported that the implementation of CAR-T treatments would require some additional investments, equal to a minimum of €15,868.63 for a medium-large size hospital, with all the requirements imposed for the accreditation by Joint Commission International (JCI). This investment would increase, becoming equal to € 102,949.49, in case of introduction of an additional hospital bed in an ICU, suggesting have a dedicated ICU bed for the management of the more severe adverse events or in case of a high-volume CAR-T patients’ management activities.
This aspect is also relevant to support the decision of a Region to deliver CAR-T cell therapy, by the accreditation of a specific hospital. In this view, the Regional decision maker should select where to invest in the treatment delivery or to send the patient requiring therapy in another Region. This topic is acquiring a strategic relevance since the Italian NHS lacks of specific indications by regional and national authorities for necessary infrastructural and organizational investments can leave CAR-T centers carrying investment costs by themselves.
The economic and organizational analysis presented may support healthcare organization since are currently facing financial sustainability challenges, that are amplified in case of insufficient per-patient funding tariff for CAR T-cell therapy procedures.
We added these considerations in the Conclusion section of the revised manuscript

This manuscript is a resubmission of an earlier submission. The following is a list of the peer review reports and author responses from that submission.
Round 1
Reviewer 1 Report
Thanks for the invitation of review, and I have some concerns shown as below.
The research contribution is unclear. What has been known in previous studies? What new finding has been avhieved in this investigation? This manucript is very difficult to follow.
The present study seems like a review article that summerizes findings in previous studies. However, the numbers (economic evaluation) provided in Table 4, 5, and 6 lack sound empirical support, literature citing, or being based on clear calculation approach from authors their own.
The key results displayed in Table 2 and 3 mainly come from TWO to THREE literatures. It is neither research finding of author themselves nor a thorough summarization of previous relevant literature.
Reviewer 2 Report
I think that this study is very interesting! However, I strongly suggest a review of the English language, as indicated in the version I have annotated - I was unable to upload it and have sent it to the journal for uploading. Sadly, when transferring a PDF to Word (so that I could easily work in track changes and add notes), the formatting is not ideal, but you'll get the gist of my recommendations.
While the study is important, given the horrendous cost of CAR-T therapies, I think that the Method and Results section would benefit from more clarity, using a more succinct style of writing.
I also suggest adding the limitation of the Italian reimbursement being: a) based on negotiated prices that are confidential; and b) that not all markets use the payment by results approach. It's unclear if you've used published list prices or the negotiated prices.
